# OpenReview forum: "Origins and roles of world representations in neural networks"
_ICLR.cc/2026/Conference — Submitted to ICLR 2026_

### Official Review · Reviewer_taAU · 2025-10-26

**Soundness:** 2
**Presentation:** 3
**Contribution:** 2
**Rating:** 2
**Confidence:** 3

**Summary:**

The present paper proposes a framework for studying representations in neural networks. In particular, the authors separate input data ("world") from task ("data generation"), and then study how training on single and multiple tasks effects the emerging representations. They find that training on single tasks yields divergent representational geometries, whereas multi-task training drives alignment. Furthermore, their results indicate that representational divergence measured in single-task pretraining predicts downstream failure during finetuning.

**Strengths:**

The paper was overall easy to follow and read. The broader question (how to make sense of neural network representations) is of both intellectual and practical interest. From a conceptual perspective, I liked the proposed seperation between data and task. In terms of results, the paper showed very cleanly that multi-task learning leads to more aligned model internal representations.

**Weaknesses:**

While the authors argue that their work represents evidence for the platonic representation hypothesis, that is at best only weakly the case. True evidence would require studying representations of different model architectures, which is not done presently.

Related work is not much discussed. I am not too familar with this line of work, but surely people have studied neural network representations in multi-task settings before. How does the present work connect to this and in which aspects does it differ?

Personally, I found most of the results not very surprising and somewhat limited in impact. While the methodology is sound and the analysis thorough, the findings largely align with my prior expectations.

The authors argue that "generalization performance correlates with the CKA values from single-task pretraining." While that is true, the relationship is fairly weak.

Limiting the analysis to seven tasks seems constraining, especially if one eventually wants to transfer the insights to realistic settings.

Minor:
* Figure labels generally very small.
* Figure 1 not referenced in the text.
* Figure 4 abbreavtions not defined.
* Figure 6 not referenced in the text.
* Figure 6 dual axis very confusing without color coding.

**Questions:**

What is the actual input to the transformer? I assume it is just the entire string? Is it the same for the single- and multi-task settings? This is never explicitly mentioned. How are strings tokenized? Everything on the character-level except for city ids?

How consistent in their representations are single-task models across mutliple runs? That seems like an important control condition.

Crossing fails to train alone. Why? That seems strange given that this is a fairly simple setup.

"Despite these differences, we can still linearly decode (x,y) coordinates from most tasks, as shown in the second row of Fig. 4." Where can I see this in the figure?

---

> ### Author Response · Authors · 2025-11-17
> **Quick questions before the main rebuttals!**
>
> Thank you for your review!
>
> Before sharing the main rebuttals, we felt it might be useful to ask a few quick questions.
>
> 1. **Architecture**: Given the time it probably is challenging to set up a RNN model which are also very slow to train. Is there an architecture you would suggest for sequence modeling tasks?
>
>
> 2. **Literature on Multi-Task Settings**: This was our prior as well!  Surprisingly, we didn't find any studies studying the representational aspect of multi task learning in a controlled setup. (We read: https://arxiv.org/abs/1707.08114)
> There seems to be some theoretical work (Maurer, 2016; Lu 2025)  and some works studying the scaling law aspects (Michaud 2023, 2025). However we didn't find any works studying the adaptation of networks to multiple tasks based on an internal representation perspective. As you suggest, we will be looking harder for similar works as much as possible, please let us know if you have suggestions!
>
>
> 3. **Limiting the analysis to seven tasks seems constraining**: Would the concern be more about the number 7 being not big enough or more generally about the setup?
>
> We are still preparing the main rebuttals/updates. Thank you for your feedback, especially on the framing, scope and presentation of the paper!

---

> > ### Comment · Reviewer_taAU · 2025-11-19
> >
> > 1. What type of architecture doesn't matter so much to me, but if you want to make claims regarding platonic representation hypothesis, different architectures would have to be studied because that is that this hypothesis is about. Or, alternatively, such claims could be removed.
> >
> > 2. Even if not 100% related, it would be useful to give examples of what you believe to be the nearest neighbor to your work.

---

> ### Author Response · Authors · 2025-11-23
>
> ## **Response to Reviewer taAU**
>
> Thank you for your review! We appreciate that you found the paper "easy to follow," the broader question "of both intellectual and practical interest," and the separation between data and task conceptually appealing. We address the weaknesses and questions below.
>
> ---
>
> ### **W1: Evidence for Platonic Representation Hypothesis**
>
> > *"While the authors argue that their work represents evidence for the platonic representation hypothesis, that is at best only weakly the case. True evidence would require studying representations of different model architectures, which is not done presently."*
>
> Thank you for this important point. We agree that cross-architecture convergence is central to PRH, and we have reframed our claims accordingly. We now present our work as partial evidence for the *Multitask Scaling Hypothesis* (which is more closely directly related), one proposed mechanism underlying PRH, rather than for PRH itself. Here are the specific changes:
>
> | Location | Original | Revised |
> |----------|----------|---------|
> | **Abstract** | "providing controlled evidence for the Platonic Representation Hypothesis" | "providing controlled evidence for the **Multitask Scaling Hypothesis, a potential explanation of** the Platonic Representation Hypothesis" |
> | **Intro bullet** | "provide controlled evidence for the Platonic Representation Hypothesis" | "provides **partial** evidence for the **Multitask Scaling Hypothesis, one proposed mechanism for** the Platonic Representation Hypothesis" |
> | **Result 3 title** | "Evidence for the Platonic Hypothesis" | "**Task diversity aligns representations**" (PRH removed from title) |
> | **Result 3 body** | "Our results speak directly to the recently proposed Platonic Representation Hypothesis" | "This question **partially connects** to the Platonic Representation Hypothesis... One potential mechanism they suggest is the Multitask Scaling Hypothesis" |
> | **Result 3** | (no caveat) | **Added footnote**: "A full test of the Platonic Representation Hypothesis would require showing convergence across different architectures; we test only the task-diversity mechanism here." |
> | **Conclusion** | "provides controlled evidence for the recently proposed Platonic Representation Hypothesis" | "providing **partial** evidence for the **Multitask Scaling Hypothesis**" |
> | **Limitations** | (not present) | **Added**: "our claims regarding the Platonic Representation Hypothesis are partial: we demonstrate task-driven convergence within a single architecture and modality, but do not explore true multimodality or cross-architecture convergence" |
>
> That said, we believe establishing a controlled, testable setup has value given the tremendous interest in PRH. The original work (Huh et al., 2024) hypothesizes the Multitask Scaling Hypothesis but does not test it experimentally or defines a setup to test it. We offer the first manipulable framework to start validating it.
>
> We also note that testing cross-architecture convergence for sequence modeling is challenging: the landscape remains dominated by transformers, and in fact all 20 language models analyzed in the original PRH paper are transformers. Our transformer-focused study thus represents a natural starting point, while cross-architecture comparisons remain important future work.
>
> We hope this alleviates your concern.
>
> **Actions Taken:**
> - Systematic revision of all PRH claims (see table above)
> - Added explicit limitation in Results section (footnote)
> - Added explicit limitation in Discussion section
>
> ---
>
> **[1/n, Continued Below]**

---

> > ### Author Response · Authors · 2025-11-23
> >
> > ---
> >
> > ### **W2: Related Works**
> >
> > > *"Related work is not much discussed. I am not too familar with this line of work, but surely people have studied neural network representations in multi-task settings before. How does the present work connect to this and in which aspects does it differ?"*
> >
> > Thank you for this feedback. We have now added a dedicated Related Work section to the main text and expanded the appendix discussion.
> >
> > We shared your intuition that "surely people have studied neural network representations in multi-task settings before." However, despite familiarity with both representational studies and synthetic data studies, we found that the interaction of multi-task learning and representational geometry is genuinely a gap in the literature. We surveyed a substantial body of work:
> >
> > - Kim et al. (2024) [1] studies ICL plateaus but not internal representation geometry
> > - Aghajanyan et al. (2021) [10] uses "representation" implicitly—they study transfer learning benefits but do not examine internal representations directly
> > - Kumar et al. (2022) [9] studies fine-tuning effects on features but not rich world representations with ground-truth structure
> > - Maurer et al. (2016) [11] provides theoretical analysis of multi-task benefits but no empirical study of representation geometry
> > - Yang & Hospedales (2017) [12] surveys multi-task learning comprehensively but focuses on architectures and optimization, not representational convergence
> >
> > The closest works we found in controlled settings are Michaud et al. (2023) [13] and Zhang et al. (2025) [14]. However, Michaud et al. studies a single task family with different parameters, while Zhang et al. mixes cellular automata rules but focuses on behavioral capabilities rather than internal representations.
> >
> > Our study was designed to fill this gap: in reality, LLMs are used for multiple tasks, and new entities (words, concepts, abstractions) must be integrated via gradient descent. Yet the science of LLMs often still studies single tasks. We hope our framework helps bridge this gap.
> >
> > We added a dedicated Related Work section in the main text:
> >
> > > *"Multi-task Learning. Multi-task learning has long been studied as a way to improve generalization through shared representations (Caruana, 1997); in some sense, modern foundation models represent an extreme form of multi-task training. Large-scale multi-task pretraining typically assumes rich representations emerge from data diversity (Aghajanyan et al., 2021), but the precise mechanisms remain underexplored. Recent work has begun studying task diversity in controlled settings (Michaud et al., 2023; Zhang et al., 2025), though most studies still focus on aggregate behaviors such as capabilities and scaling laws rather than characterizing tasks or the knowledge they operate on. Our framework explicitly defines tasks as geometric functions over a shared world, enabling direct investigation of how task structure shapes representations."*
> >
> > **Actions taken:**
> > - Added Related Work section to main text
> > - Expanded related works in appendix with comprehensive survey
> >
> > ---
> >
> > **[2/n, Continued Below]**

---

> ### Author Response · Authors · 2025-11-23
>
> ---
>
> ### **W3: Surprisingness of the Results**
>
> > *"Personally, I found most of the results not very surprising and somewhat limited in impact. While the methodology is sound and the analysis thorough, the findings largely align with my prior expectations."*
>
> We agree that findings up to Section 3 may be unsurprising to some researchers, **in fact including us**. However, we find that many researchers still find the existence of non-superficial representations by itself surprising (this is only our result 1), as evidenced by the attention given to work on world models in language models [2], geometric structure in truth representations [3], and monosemantic features in neural networks [4,5]. This divergence of expectations motivated our study: we aimed to establish a controlled framework reproducing the emergence of world representations, to exactly bring together the audience which believes it is surprising and the ones which think it is not.
>
> While Section 3 may align with some expectations (including yours and ours), it provides controlled experimental validation showing that: (1) world representations robustly emerge via next-token prediction across many tasks, (2) different tasks give rise to different representations despite surfacing similar linear readouts, and (3) multi-task training drives representational convergence.
>
> However, we believe Section 4's findings are genuinely surprising. We expected joint training on 7 tasks to create shared representations that would benefit from any single-task fine-tuning. Instead, we discovered that the distance task actively harms representational integration, causing new entities to be encoded in hidden subspaces and degrading performance on all other tasks. Looking at our 7 geometric tasks, we see no obvious reason why distance would be problematic. If you found these Section 4 results also unsurprising, we would genuinely like to understand your perspective.
>
> ---
>
> ### **W4: Weak CKA-Generalization Correlation**
>
> > *"The authors argue that "generalization performance correlates with the CKA values from single-task pretraining." While that is true, the relationship is fairly weak."*
>
> We agree the relationship is weak, which is why we carefully hedge this claim throughout the manuscript. Running multiple seeds did strengthen the correlation somewhat, but we still consider it a weak relationship. The surprising aspect isn't the correlation's strength, but that *any* relationship exists between single-task properties and multi-task fine-tuning dynamics: these are completely different training settings. We present this as an intriguing observation for future investigation, not a central claim.
>
> We have revised Section 4's title to focus on the more robust finding: divergent tasks harming fine-tuning generalization. The CKA correlation is now presented as a secondary observation that may help identify such tasks. We also strengthened the divergent task and misintegration findings with additional quantification (linear probe reconstruction error as an integration metric).
>
> **Actions taken:**
> - Revised Section 4 title to focus on divergent tasks
> - Strengthened divergent task analysis with additional quantification
> - Clarified tentative nature of CKA-generalization correlation
> - Multi-seed experiments strengthen quantitative reliability (correlation improved but still weak)
>
> ---
>
> ### **W5: Limited Number of Tasks**
>
> > *"Limiting the analysis to seven tasks seems constraining, especially if one eventually wants to transfer the insights to realistic settings."*
>
> We acknowledge this limitation. However, we chose 7 tasks deliberately: this enables exhaustive analysis of all pair and triple combinations (C(7,2)=21 and C(7,3)=35) with statistical power while remaining tractable. Importantly, we selected *qualitatively* different geometric computations, likely requiring different internal circuits, rather than parameter variants of the same task family, which is more common in the literature.
>
> We agree that scaling to more tasks would enable systematic study of how task count affects convergence dynamics. However, we don't think that simply adding tasks would make the study more valuable or more insightful for practical models. We believe the key contributions are the framework itself and the discovery of divergent tasks, both of which can be extended in future work.
>
> ---
>
> **[3/n, Continued Below]**

---

> > ### Author Response · Authors · 2025-11-23
> >
> > ### **Minor Points**
> >
> > > *"Figure labels generally very small. Figure 1 not referenced in the text. Figure 4 abbreviations not defined. Figure 6 not referenced in the text. Figure 6 dual axis very confusing without color coding."*
> >
> > Thank you for catching these issues! All have been addressed:
> > - Enlarged figure labels throughout
> > - Added Figure 1 now referenced in text
> > - Added abbreviation legend to Figure 4 (Now Fig 2)
> > - Figure 6 (Now Fig 4) is now referenced
> > - Added dotted lines to right axis using tasks
> >
> > Thank you again for catching these presentation issues!
> >
> > ---
> >
> > ### **Q1: Transformer Input and Tokenization**
> >
> > > *"What is the actual input to the transformer? I assume it is just the entire string? Is it the same for the single- and multi-task settings? This is never explicitly mentioned. How are strings tokenized? Everything on the character-level except for city ids?"*
> >
> > You are correct: the entire task string is fed to the transformer as input. For example, `dist(c_0865,c_4879)=769` becomes the character sequence `<bos> d i s t ( c _ 0 8 6 5 , c _ 4 8 7 9 ) = 7 6 9 <eos>`. This is identical for both single-task and multi-task settings—the only difference is the diversity of task types in the training data.
> >
> > We intentionally use character-level tokenization throughout (including city IDs and numerical outputs) to present the transformer with challenges analogous to real LLMs: multi-token entities, arithmetic on broken numbers, and computations at varying sequence positions. This ensures our findings aren't artifacts of simplified tokenization.
> >
> > We apologize for this omission and have now documented these details in **App. B**, including the vocabulary size (98 ASCII tokens) and formatting conventions.
> >
> > **Actions taken:**
> > - Expanded App. B with detailed tokenization and input format documentation
> >
> > ---
> >
> > ### **Q2: Cross-Seed Consistency**
> >
> > > *"How consistent in their representations are single-task models across multiple runs? That seems like an important control condition."*
> >
> > Thank you for this question. We note that single-task CKA consistency isn't central to our main claims, but reproducibility across seeds certainly matters.
> >
> > We trained models with multiple seeds: 3 seeds for all pretraining configurations and 4 seeds for fine-tuning. Results now include mean ± SEM in the Appendix.
> >
> > To clarify: we do not say that visually similar representations imply high CKA. In fact, Section 3 notes the opposite: *"This reveals that the distance task produces significantly different model representations—a result not expected intuitively."* Indeed, same-task-different-seed CKA can be quite low, especially for distance.
> >
> > Our key findings are robust across seeds:
> >
> > - Distance consistently emerges as divergent (**Fig. 2b, App. Fig. 11**)
> > - CKA increases with task count (1→2→3 tasks) across all seeds (**Fig. 3d, App. Fig. 13, 14a**)
> > - Fine-tuning patterns replicate with mean ± SEM across 4 seeds (**Fig. 5, 6, App. Fig. 15–18**)
> > - CKA-generalization correlation improved with averaging (R²: 0.126 → 0.188), indicating single-seed estimates were noisy (**Fig. 5b**)
> >
> > **Actions taken:**
> > - 3-seed repeats for all pretraining experiments; 4-seed repeats for all fine-tuning experiments
> > - All results now report mean ± SEM; individual seed results shown in Appendix
> > - Added analysis of cross-seed CKA variance as function of task count
> >
> > ### **Q3: Crossing Task Failure**
> >
> > > *"Crossing fails to train alone. Why? That seems strange given that this is a fairly simple setup."*
> >
> > This is not a bug: from our experience training transformers on synthetic data, single-task optimization can be surprisingly challenging. It may be counterintuitive that models which train successfully on complex language struggle with simple small tasks, but this is indeed the case! Loss plateaus are a well-known phenomenon [6,7,8], and gradient-based methods can fail on deceptively simple problems [15]. Binary classification tasks like crossing are particularly prone to getting stuck. Transformers have been shown to fail on seemingly simple tasks like parity or short-range lookback in isolation [16].
> >
> > Importantly, crossing succeeds when combined with *any* other task. We speculate that companion tasks provide structured coordinate representations that crossing can leverage, i.e. an implicit curriculum.
> >
> > **Actions taken:**
> > - Added discussion of loss plateaus to Related Work, as this may not be intuitive and could appear like a bug
> >
> > ---
> >
> > ### **Q4: Linear Decoding in Figure**
> >
> > > *"'Despite these differences, we can still linearly decode (x,y) coordinates from most tasks, as shown in the second row of Fig. 4.' Where can I see this in the figure?"*
> >
> > This is visible in the second row of **Fig. 2a** (previously Fig. 4a), where the two dimensions within the plane are the linear probe directions for x and y coordinates. We have also added quantitative probe R-square results in **App. Fig. 8**.
> >
> > ---
> >
> > **[4/n, Continued Below]**

---

> > > ### Author Response · Authors · 2025-11-23
> > >
> > > **Summary**
> > >
> > > Thank you again for the constructive feedback! Your comments on framing and calibration were particularly helpful.
> > >
> > > Also thank you for reading the long rebuttal!
> > >
> > > We hope these extensive revisions and new experiments address your concerns. If so, we would be grateful if you could consider increasing your score.
> > >
> > > We are happy to discuss any remaining questions or concerns!
> > >
> > > ---
> > >
> > > **References:**
> > >
> > > [1] Kim et al. (2024): https://arxiv.org/abs/2410.05448
> > >
> > > [2] Gurnee & Tegmark (2023): https://arxiv.org/abs/2310.02207
> > >
> > > [3] Marks & Tegmark (2024): https://arxiv.org/abs/2310.06824
> > >
> > > [4] Bricken et al. (2023): https://transformer-circuits.pub/2023/monosemantic-features/index.html
> > >
> > > [5] Templeton et al. (2024): https://transformer-circuits.pub/2024/scaling-monosemanticity/
> > >
> > > [6] Pezeshki et al. (2021): https://arxiv.org/abs/2011.09468
> > >
> > > [7] Shah et al. (2020): https://arxiv.org/abs/2006.07710
> > >
> > > [8] Hoffmann et al. (2024): https://arxiv.org/abs/2310.12956
> > >
> > > [9] Kumar et al. (2022): https://arxiv.org/abs/2202.10054
> > >
> > > [10] Aghajanyan et al. (2021): https://arxiv.org/abs/2101.11038
> > >
> > > [11] Maurer et al. (2016): https://arxiv.org/abs/1505.06279
> > >
> > > [12] Yang & Hospedales (2017): https://arxiv.org/abs/1707.08114
> > >
> > > [13] Michaud et al. (2023): https://arxiv.org/abs/2303.13506
> > >
> > > [14] Zhang et al. (2025): https://arxiv.org/abs/2410.02536
> > >
> > > [15] Shalev-Shwartz et al. (2017): https://arxiv.org/abs/1703.07950
> > >
> > > [16] Bachmann et al. (2024): https://arxiv.org/abs/2403.06963

---

> > > > ### Comment · Reviewer_taAU · 2025-11-24
> > > >
> > > > Thanks for the response. I have adjusted my score accordingly.

---

> > > > > ### Author Response · Authors · 2025-11-28
> > > > >
> > > > > We thank the reviewer again for their thorough review!
> > > > >
> > > > > ---
> > > > >
> > > > > It would genuinely help if you could describe why the Section 5 results (fine tuning misintegration) might be unsurprising from your perspective, but if its an intuition which is hard to articulate, we totally understand !!
> > > > >
> > > > > ---
> > > > >
> > > > > Thank you again, and let us know if you have any more questions.

---

### Official Review · Reviewer_oLe1 · 2025-10-29

**Soundness:** 3
**Presentation:** 2
**Contribution:** 2
**Rating:** 6
**Confidence:** 3

**Summary:**

The authors present a study investigating how neural networks (transformers specifically) learn convergent representations of a single latent data manifold through different tasks, or combinations thereof. They use real world cities to define a set of latent coordinates, and come up with seven function learning tasks, mapping city tuples to outputs that the networks are trained to predict. The authors show that, when trained on single tasks, the models learn representations that tend to by similar (measured by CKA), but also show structural differences. By constraining the representation space by training the models on more tasks, the representations start to align more. This nicely demonstrates a principle often formalized as the Platonic Representation Hypothesis. Lastly, the authors analyze representations when models are fine-tuned to incorporate a novel (fictitious) city.

**Strengths:**

* The authors conduct simple and diagnostic experiments. The results make sense and support the conclusions made in the paper
* Nice diagnostic test of the platonic representation hypothesis in a toy setting

**Weaknesses:**

* The introduction meanders on very general questions related to representation learning and neural networks, not easy to see how all of these are related to the questions the paper actually studies. I would recommend making the introduction more succinct and to-the-point.

Overall, the paper provides good evidence for a simple and interesting question. While it's not very surprising that multi-task training constrains the model representations, giving rise to alignment, the evidence presented is solid so I'm happy to recommend accept.

Formatting:
* On line 383 there seems to be a reference missing (see the question mark)

**Questions:**

* Were other model training factors tested? For instance, regularization (L1 or L2) on the residual stream representations might speed up alignment, as constraints are put on the representations.
* Does multi-task alignment interact with model size? Were differently sized transformers trained?

---

> ### Author Response · Authors · 2025-11-23
>
> ## **Response to Reviewer oLe1**
>
> Thank you for your review! We are glad you found our experiments "simple and diagnostic" with "solid evidence," and appreciate the recommendation to accept! We address the weaknesses and questions below.
>
> ---
>
> ### **W1: Introduction**
>
> > *"The introduction meanders on very general questions related to representation learning and neural networks, not easy to see how all of these are related to the questions the paper actually studies. I would recommend making the introduction more succinct and to-the-point."*
>
> Thank you for this feedback. We agree the original introduction might have been too broad. We have revised it by:
> - Removing the long paragraph discussing LLMs.
> - Removing speculative questions about disentanglement, alternative learning algorithms, etc.
> - Creating a more direct path: open questions → need for controlled setup → our framework → contributions
> - Moving detailed literature discussion to a dedicated Related Work section
>
> The introduction now focuses on what we actually study rather than broad context.
>
> Thank you very much for this comment, it likely is much easier for the reader to follow now!
>
> **Actions taken:**
> - Revised introduction
>
> ---
>
> ### **On Surprisingness**
>
> > *"While it's not very surprising that multi-task training constrains the model representations, giving rise to alignment, the evidence presented is solid so I'm happy to recommend accept."*
>
> We appreciate this recognition. We agree that multi-task convergence aligns with expectations from the Platonic Representation Hypothesis (Huh et al., 2024). However, we believe providing a controlled, experimentally testable system adds value. It allows asking precise follow-up questions: Which tasks drive convergence? How does seed variance interact with task diversity? Are Platonic-Converged representation always "better" ?
>
> Most importantly, we highlight the discovery of "divergent tasks" (Section 4): despite joint multi-task pretraining, certain tasks (like distance) actively harm representational integration during fine-tuning. This challenges the assumption that multi-task training produces unified representations by showing that some tasks can use hidden representation spaces which manifest during adaptation. We believe this finding is less expected and has implications for understanding when and why fine-tuning fails in practice. In fact, we didn't expect it at all.
>
> ---
>
> ### **Formatting**
>
> > *"On line 383 there seems to be a reference missing (see the question mark)"*
>
> Fixed, thank you!
>
> **[1/n, Continued Below]**

---

> ### Author Response · Authors · 2025-11-23
>
> ---
>
> ### **Q1: Other Training Factors (Regularization)**
>
> > *"Were other model training factors tested? For instance, regularization (L1 or L2) on the residual stream representations might speed up alignment, as constraints are put on the representations."*
>
> We explored weight decay and learning rate variations during development. These primarily affected loss plateau dynamics (consistent with literature on gradient plateaus in transformers [1,2]) but did not significantly alter the final representational geometry.
>
> Could you clarify what you mean by "speeding up alignment"? Would you mean as a function of training steps?
>
> If so, interestingly our analysis reveals that representations largely stabilize after the initial sharp loss drop; once formed, they exhibit minimal movement throughout the rest of training (**App. Fig. 16**). This early stabilization *potentially* suggests the final global geometry is determined relatively quickly, leaving limited room for regularization to "speed up" alignment. Whether regularization could change *which* geometry forms is an interesting direction we have not explored.
>
> ---
>
> ### **Q2: Model Size**
>
> > *"Does multi-task alignment interact with model size? Were differently sized transformers trained?"*
>
> This is an interesting question! Currently, we do not know how multi-task alignment interacts with model size during pretraining. Our intuition from training many models during the research process is that it should not change substantially, but this remains an empirical question.
>
> However, motivated by your question, we at least wanted to confirm whether the divergent task phenomenon might be solved by scale. We trained a 2× wider transformer (256 hidden dim, 1024 intermediate, 8 heads vs. our baseline 128/512/4). The key finding: **the divergent task pattern persists**. Distance-containing fine-tuning combinations still harm cross-task generalization even with increased capacity (**App. Fig. 20**). This provides some evidence that it is not simply a model capacity problem.
>
> We agree that a more systematic study across model scales, with careful consideration of capacity, scaling laws, and training dynamics, would be valuable future work.
>
> ---
>
> **Summary**
>
> Thank you again for the helpful feedback! We hope these responses address your concerns, and would be glad if you could recommend our paper more strongly.
>
> Happy to discuss further or answer any questions!
>
> ---
>
> **References:**
>
> [1] Pezeshki et al. (2021): https://arxiv.org/abs/2011.09468
>
> [2] Shah et al. (2020): https://arxiv.org/abs/2006.07710

---

> > ### Author Response · Authors · 2025-11-28
> >
> > Given the discussions will close soon, may I ask you reviewer oLe1, if your main concerns are well addressed?
> >
> > Thank you for the revision again!

---

### Official Review · Reviewer_8dPS · 2025-10-31

**Soundness:** 2
**Presentation:** 2
**Contribution:** 3
**Rating:** 4
**Confidence:** 3

**Summary:**

The paper introduces a simple framework to analyze how different training objectives influence the learned representations when the underlying world model is known. To make this analysis tractable, the authors construct a synthetic setup where the “world” consists of 2D city coordinates. Data are generated using seven different geometric tasks based on these coordinates. This controlled environment allows the authors to systematically study several aspects of representation learning: First, they show that models trained on most of these tasks can learn representations that are linearly mappable to the underlying 2D coordinates, thereby capturing the correct world model. Second, they demonstrate that combining multiple tasks improves this alignment with the true world representation; for some tasks, such multi-task training is even necessary for success. Lastly, they investigate fine-tuning effects: when a general-purpose model is fine-tuned on a subset of tasks with additional data, the choice of fine-tuning task is crucial. In some cases, new “cities” embed seamlessly into the existing latent geometry, while in others they occupy a separate region of the space. This behavior also determines whether the fine-tuned model generalizes across tasks or becomes specialized to the task it was fine-tuned on

**Strengths:**

The paper presents a well-designed framework for systematically analyzing how training objectives affect learned representations when the underlying world model is known. The idea of generating tasks based on a shared, low-dimensional representation is elegant and enables controlled, interpretable experiments. The seven tasks are well-chosen and require different forms of geometric reasoning. The overall problem setup is clearly explained, and Figure 2 effectively illustrates the environment and task construction. By isolating the training objective from other factors such as architecture or data complexity, the findings become easier to interpret.
The visualization in Figure 3 provides an intuitive view of how the world model emerges during training, demonstrating that this emergence is not necessarily directly correlated with task performance. Overall, the proposed framework serves as a valuable analytical tool for studying and bench-marking representation learning methods.

**Weaknesses:**

- The study appears to rely on a single random seed for training. It would be important to evaluate whether the learned representations vary more across different tasks than across different random initializations of the same task. Without such analysis, it is difficult to assess the stability of the reported findings.
- The character-based city encoding is somewhat unconventional and insufficiently justified. It is unclear how the cities are indexed or numbered—if the numbering follows geographic order, this could inadvertently leak coordinate information into the model. Furthermore, the authors note in the appendix that city coordinates starting with “0”, “00”, or “000” fail to work and were excluded from all experiments. This exclusion raises concerns about potential implementation artifacts or biases in the input representation.
- In the fine-tuning experiments, the new coordinates are concentrated in a small region of the space, leading to clustering of new cities. This represents a special case of localized additional information, while experiments using more spatially distributed new points would help evaluate whether the conclusions generalize.
- The city manifold used in this framework is a flat 2D plane, while real-world data often lie on more complex manifolds. Extending the approach to non-Euclidean geometries, such as a spherical globe, would be an interesting next step and could test the robustness of the proposed framework.
- The paper provides only a limited explanation of what constitutes “divergent tasks.” A deeper discussion of the specific geometric reasoning required by each task and why certain tasks diverge would help clarify this concept.
- Minor comments:
   - The caption of Figure 3 mentions “top,” “middle,” and “bottom” panels, but the figure appears as a single mixed layout.
   - The caption and description of Figure 5 refer to a 21×21 CKA matrix, while the plot shows only 7×7.

**Questions:**

- As the Crossing task did not succeed on its own, did it work in combination with any other tasks, or only with Distance and Perimeter? In addition, while Distance appears to perform well in early experiments, it is later described as a divergent task. Could this divergent behavior be an artifact of the fine-tuning data, where the new cities (“Atlantis”) are concentrated in a single coordinate region?
- What are the accuracies of the linear probe used for coordinate prediction across different settings—models trained on single tasks, combined tasks, and the fine-tuned variants? Including these results would help quantify how well each representation captures the true world geometry.
- How is Normalized Improvement defined? Please clarify how it is calculated. Also, it is unclear, whether the deviation from max-model is in percentage or absolute units.

---

> ### Author Response · Authors · 2025-11-17
> **Quick clarification before the main rebuttals!**
>
> Thank you for your through review!!
>
> Motivated by your suggestions, we are currently preparing a set of additional experiments and analysis.
>
> Before we share the main rebuttal, we have a clarification question:
>
> **"The study appears to rely on a single random seed for training."**: Are you more concerned about the neural nets' initialization seed or the random ordering of cities? Does the latter mean that there is only one setup (i.e. some kind of "implicit seed")? We could definitively try to source the "world" data differently, but probably won't have time to replicate the whole set of experiments. Would there be a main result you would like to see robustified?
>
> Thank you so much for your review, your review is genuinely helpful and we will be in touch soon!

---

> > ### Comment · Reviewer_8dPS · 2025-11-18
> > **Answer: Quick clarification before the main rebuttals!**
> >
> > Thank you for the clarification.
> > My concern is specifically about how large the variation in learned representations is across different random seeds of the same task, compared to the variation between different tasks. This would help contextualize Figures 4c and 5c: if representation differences caused by randomness are comparable to those caused by task choice, the interpretation of those figures would change.
> > A related metric that would help here is the linear-probe accuracy (with standard deviations) for recovering x,y coordinates across tasks, as it would directly reflect representation stability.
> >
> > A simple way to address this would be to train a few additional models (even 2–3) per task with different network initialization seeds and random city encodings and report the variability. The focus could be limited to the Section 3 analysis, so that a full re-run of all experiments would not be necessary.

---

> ### Author Response · Authors · 2025-11-23
>
> ## **Response to Reviewer 8dPS**
>
> Thank you for your thorough review. We are glad you found the framework "well-designed" and "elegant," and that the seven tasks are "well-chosen." We put considerable effort into cleanly separating world structure from data generation process and are glad to hear this came through! Below we address each weakness and question.
>
> ---
>
> ### **W1: Single Seed**
>
> > *"The study appears to rely on a single random seed for training. It would be important to evaluate whether the learned representations vary more across different tasks than across different random initializations of the same task. Without such analysis, it is difficult to assess the stability of the reported findings."*
>
> Thank you for raising this important point. We have now run multiple seeds for all experiments: 3 seeds for pretraining (single-task, 2-task, 3-task, and 7-task) and 4 seeds for fine-tuning. All plots and metrics now report mean ± SEM (in Appendix), and show single seed results for CKA vs number of tasks.
>
> Before presenting the results, we want to clarify the original claim (just in case): **we did not suggest that qualitative visual similarity implies high CKA.** In fact, we noted the opposite in Section 2 (now section 3): *"This reveals that the distance task produces significantly different model representations—a result not expected intuitively."* The multi-seed analysis confirms this: same-task-different-seed CKA can indeed be quite low, especially for the distance task! (We should probably have masked out the diagonal since we were running single seed.)
>
> In any case, we ran 3 seeds for all 21 (7 1task, 7 2task, 7 3task) models, total 63 runs. Overall, our findings replicates and is in fact reinforced!
>
> We observe:
> - The distance task consistently stands out as divergent across all seeds, and thus on average as well. (**Fig. 2b, App. Fig. 11**)
> - CKA trends (1→2→3 task convergence) hold when averaged across seeds. (**Fig. 3d, App. Fig. 13, App. Fig. 14a**)
> - Fine-tuning generalization patterns are robust (now reported as mean ± SEM across 4 seeds) (**Fig. 5,6, App. Fig. 15,16,17,18**)
> - The CKA-to-generalization correlation actually *improved* with seed-averaging (R²: 0.126 → 0.188), suggesting single-seed measurements were noisy estimates of the real underlying relationship (**Fig. 5b**)
>
> **Additionally:** Multi-task learning not only drives cross-task representational convergence but also **reduces cross-seed variance** when selecting same tasks (**App. Fig. 14b**). Single-task models show high sensitivity to initialization, while multi-task models converge to more consistent representations regardless of seed.
>
> **Actions taken:**
> - 3-seed repeats for all pretraining experiments; 4-seed repeats for all fine-tuning experiments
> - All results now report mean ± SEM; individual seed results shown in Appendix
> - Added analysis of cross-seed CKA variance as function of task count
> - 8 Figures added.
>
> ---
>
> **[1/n, Continued Below]**

---

> > ### Author Response · Authors · 2025-11-23
> >
> > ---
> >
> > ### **W2: City ID Assignment**
> >
> > > *"The character-based city encoding is somewhat unconventional and insufficiently justified. It is unclear how the cities are indexed or numbered—if the numbering follows geographic order, this could inadvertently leak coordinate information into the model. Furthermore, the authors note in the appendix that city coordinates starting with "0", "00", or "000" fail to work and were excluded from all experiments. This exclusion raises concerns about potential implementation artifacts or biases in the input representation."*
> >
> > This is a reasonable concern—let us discuss this in detail (we have now added a dedicated discussion in **App. B.3**).
> >
> > **On tokenization:** We intentionally chose character-level tokenization rather than single-token-per-entity encoding. Our preliminary experiments reproducing pitfalls of next-token prediction [1] showed that tokenization details significantly change synthetic experimental results. Real LLMs must handle multi-token entities, variable-length prompts, different task formats, and broken numerical tokens, computation at different positions, etc. We chose character-level encoding to better approximate these realistic challenges rather than using a synthetic-friendly scheme where each entity is a single token. Furthermore, if we tokenize each city as a token, new atlantis cities will be untrained new tokens, which is also an interesting setup but the analogy to real LLMs becomes more like soft prompts [7]. Here, we intentionally studied a setup more like real llms where all tokens are already trained and new "entities" are just new combinations.
> >
> > **On city IDs:** City IDs are **randomly assigned** with no geographic information. this is now clarified in the appendix. The exclusion of cities with IDs starting with "0", "00", or "000" is not a bug. We know exactly why it happens: the digit 0 never appears as a leading character in numerical outputs (distances, angles, etc.), so the model learns to use leading zeros as a positional signal. We exclude these cities for cleaner analysis.
> >
> > **Actions taken:**
> > - Clarified tokenization rationale and city ID assignment in appendix
> >
> > ---
> >
> > ### **W3: Clustered Atlantis Cities**
> >
> > > *"In the fine-tuning experiments, the new coordinates are concentrated in a small region of the space, leading to clustering of new cities. This represents a special case of localized additional information, while experiments using more spatially distributed new points would help evaluate whether the conclusions generalize."*
> >
> > This is a fair point. A priori, we do not expect the clustering to matter: the model must compute geometric relationships across the entire world during both pretraining and fine-tuning, and there is no explicit marker distinguishing Atlantis cities from others, its just another random id, not clustered in id space either.
> >
> > Importantly, our **pretraining-with-Atlantis experiment** (**Fig. 6d**, green line) demonstrates that Atlantis cities integrate perfectly when included from the start: Reconstruction error falls within the distribution of regular cities. This confirms that Atlantis does not possess any inherently problematic geometric properties; the integration failure is specific to the fine-tuning process, not Atlantis's location.
> >
> > However, this indeed is an experimental question we cannot be 100% sure without testing. We are currently setting this experiment up, but honestly unsure when we can get it. How essential would this be?
> >
> > **Actions taken:**
> > - Added "Atlantis in pretraining" experiment showing successful integration when included from the start
> >
> > **[2/n, Continued Below]**

---

> > > ### Author Response · Authors · 2025-11-23
> > >
> > > ---
> > >
> > > ### **W4: 2D Planar World**
> > >
> > > > *"The city manifold used in this framework is a flat 2D plane, while real-world data often lie on more complex manifolds. Extending the approach to non-Euclidean geometries, such as a spherical globe, would be an interesting next step and could test the robustness of the proposed framework."*
> > >
> > > We indeed thought very carefully about this choice, and was potentially even considering this direction of research: Our early experiments actually used spherical coordinates (longitude, latitude). However, since neural networks are highly nonlinear, there is no canonical reason why any particular external geometry (planar, spherical, hyperbolic) would be privileged by the model's internal representations. The model constructs its own representational geometry starting from random initialization.
> > >
> > > That being said, we definitively are aware that there is a subfield "geometric deep learning" which specifically studies the interaction between data geometry and neural computation. However, our focus is on general sequence modeling. Nevertheless, we have added discussion of this literature to the related works in appendix.
> > >
> > > We chose a planar world primarily for **analysis clarity** for questions we would be asking: linear probing provides a clean way to assess whether world structure is captured, whereas extracting nonlinear features is less straightforward (see Engels & Tegmark [2] and Csordás et al. [3] on challenges of reading out nonlinear representations).
> > >
> > > We totally agree that studying the underlying geometry and how concepts would organize depending on the underlying geometry is a very interesting future direction.
> > >
> > > **Actions taken:**
> > > - Added geometric deep learning discussion to related works
> > >
> > > ---
> > >
> > > ### **W5: Limited Explanation of Divergent Tasks**
> > >
> > > > *"The paper provides only a limited explanation of what constitutes "divergent tasks." A deeper discussion of the specific geometric reasoning required by each task and why certain tasks diverge would help clarify this concept."*
> > >
> > > We appreciate this question, and we must be honest: **we don't fully understand why** the distance task is divergent. Our prior expectation was that multi-task pretraining would construct shared representations that all tasks use equally. We were surprised to find that some tasks harm integration during fine-tuning, and even more surprised that this was unpredictable from task definitions alone.
> > >
> > > We initially hypothesized that distance might be "simplest" since it requires only two city inputs, but compass also uses two inputs and is arguably simpler (8-way classification vs. precise prediction). The divergence, very honestly, remains an open question.
> > >
> > > In this paper, we focused on **clearly demonstrating the phenomenon** rather than speculating about causes. We believe establishing the existence of divergent tasks is itself a contribution, and understanding the underlying cause and mechanism is important future work.
> > >
> > > We have some hypotheses, none of which we think it's worth articulating at this stage.
> > >
> > > ---
> > >
> > > ### **Minor Comments**
> > >
> > > > *"The caption of Figure 3 mentions 'top,' 'middle,' and 'bottom' panels, but the figure appears as a single mixed layout."*
> > >
> > > Thank you! Fixed: We merged Figures 1, 2, and 3 into a single overview/framework figure.
> > >
> > > > *"The caption and description of Figure 5 refer to a 21×21 CKA matrix, while the plot shows only 7×7."*
> > >
> > > We intended to say 21 elements (since there used to be 21 independent pair comparisons in the 7×7 matrix), but this analysis is supposed to be 7×7. Sorry for the confusion—we have improved the caption!
> > >
> > > ---
> > >
> > > **[3/n, Continued Below]**

---

> > > > ### Author Response · Authors · 2025-11-23
> > > >
> > > > ---
> > > >
> > > > ### **Q1: Crossing Task and Distance as Divergent**
> > > >
> > > > > *"As the Crossing task did not succeed on its own, did it work in combination with any other tasks, or only with Distance and Perimeter? In addition, while Distance appears to perform well in early experiments, it is later described as a divergent task. Could this divergent behavior be an artifact of the fine-tuning data, where the new cities ("Atlantis") are concentrated in a single coordinate region?"*
> > > >
> > > > **On crossing:** The crossing task succeeds when combined with *any* other task, not just distance or perimeter. This is consistent with known optimization challenges in transformer training on synthetic tasks. binary prediction tasks like parity and simple lookahead often fail to escape loss plateaus when trained alone [4, 5, 6]. We have added a dedicated appendix paragraph on loss plateaus and how multi-task training alleviates them.
> > > >
> > > > **On distance as divergent:** To clarify, "divergent" is a term we introduce in this paper. The distance task trains successfully and achieves low prediction error (see training dynamics in **App. Fig. 8**). By "divergent" we refer to its pretraining induced representational geometry being misaligned with other tasks, not to training failure. We do not believe the concentrated Atlantis cities selectively affect the distance task, since all tasks involve geometric calculations that depend heavily on point locations. As mentioned in W3, we are setting up a scattered Atlantis experiment to test this directly.
> > > >
> > > > **Actions taken:**
> > > > - Added appendix paragraph on loss plateaus and multi-task training
> > > > - Added training dynamics figure showing distance task trains successfully
> > > >
> > > > ---
> > > >
> > > > ### **Q2: Linear Probe Accuracies**
> > > >
> > > > > *"What are the accuracies of the linear probe used for coordinate prediction across different settings—models trained on single tasks, combined tasks, and the fine-tuned variants? Including these results would help quantify how well each representation captures the true world geometry."*
> > > >
> > > > Thank you for this  suggestion! We now provide comprehensive linear probe results:
> > > >
> > > > - **During pretraining:** We show coordinate R² across training for all models in **App. Fig. 8** (training dynamics). Linear probes achieve high R-square values for all tasks which escape the loss plateau successfully, except the compass task.
> > > > - **For Atlantis integration:** The situation is more nuanced. **Fig. 6b** shows linear probe reconstructions intuitively, and **Fig. 6d** quantifies reconstruction error. When probes are trained *without* Atlantis cities, models fine-tuned with divergent tasks show ~5× higher Atlantis reconstruction error than non-divergent task models.
> > > >
> > > > ---
> > > >
> > > > ### **Q3: Normalized Improvement Definition**
> > > >
> > > > > *"How is Normalized Improvement defined? Please clarify how it is calculated. Also, it is unclear, whether the deviation from max-model is in percentage or absolute units."*
> > > >
> > > > We have clarified the definition in the appendix. Repeated here, normalized improvement scales performance to [0, 1] where:
> > > > - **0** = The performance is same as an Atlantis city before the model sees any atlantis city.
> > > > - **1** = The performance is same as a pretrained city in the original world.
> > > >
> > > > For error-based metrics (distance, triangle area, angle, perimeter), we use log-ratio normalization:
> > > > ```
> > > > Normalized = (log(error_atlantis) - log(error_model)) / (log(error_atlantis) - log(error_standard))
> > > > ```
> > > >
> > > > For accuracy-based metrics (crossing, inside, compass), we use linear normalization:
> > > > ```
> > > > Normalized = (acc_model - acc_atlantis) / (acc_standard - acc_atlantis)
> > > > ```
> > > >
> > > > Note: Values can exceed 1 if an Atlantis city happens to perform better than the average pretrained city.
> > > >
> > > > The "deviation from max-model" in **Fig. 6a** operates in this normalized improvement space and shows the difference from the best single-task fine-tuning baseline.
> > > >
> > > > ---
> > > >
> > > > **Summary**
> > > >
> > > > We genuinely thank you for pushing us on multi-seed experiments—this turned out to be one of the most valuable suggestions we received. Our paper is now significantly stronger because of this feedback.
> > > >
> > > > We are also preparing the "scattered Atlantis" experiment and will include results if ready during the discussion period.
> > > >
> > > > We hope our changes address your concerns, and would be glad if our paper can be recommended more strongly. We are happy to discuss further or address any remaining questions.
> > > >
> > > >
> > > >
> > > > ---
> > > >
> > > > **References:**
> > > >
> > > > [1] Bachmann et al. (2024): https://arxiv.org/abs/2403.06963
> > > >
> > > > [2] Engels & Tegmark (2024): https://arxiv.org/abs/2405.14860
> > > >
> > > > [3] Csordás et al. (2024): https://arxiv.org/abs/2408.10920
> > > >
> > > > [4] Hoffmann et al. (2023): https://arxiv.org/abs/2310.12956
> > > >
> > > > [5] Gopalani et al. (2025): https://arxiv.org/abs/2506.13688
> > > >
> > > > [6] Kim et al. (2024): https://arxiv.org/abs/2410.05448
> > > >
> > > > [7] Lester et al. (2021): https://arxiv.org/abs/2104.08691

---

> > > > > ### Comment · Reviewer_8dPS · 2025-11-27
> > > > >
> > > > > We thank the authors for the very thorough and thoughtful rebuttal, as well as for the substantial additional experiments. These additions meaningfully strengthen the paper.
> > > > >
> > > > > **W1:**
> > > > > The multi-seed analysis and the additional figures are highly appreciated. One clarification question: when computing CKA across seeds, do you evaluate all cross-seed combinations (e.g., for 3 seeds and 2 models, 9 pairwise comparisons), or only same-seed comparisons (3 values)?
> > > > > The former would be preferable, since each seed induces independent training trajectories, and full cross-pairing would provide a more robust estimate of representational variability.
> > > > >
> > > > > **W2:**
> > > > > Thank you for the expanded explanation of the tokenization and ID assignment choices. The rationale for character-level encoding is now much clearer. Regarding the exclusion of IDs starting with leading zeros: I still believe that assigning IDs from a restricted range (e.g.,$[1000,9999]$) would have been a cleaner design choice, though this is a minor point and does not materially affect the contribution. It could simply be adjusted in the released code.
> > > > >
> > > > > **W3:**
> > > > > I agree that the pretraining-with-Atlantis experiment is reassuring, but the scattered-Atlantis variant would indeed provide valuable additional insight into the fine-tuning dynamics. While I understand that time constraints may limit your ability to run it during the discussion period, this would be a meaningful extension for future work.
> > > > > As a conceptual remark: in many realistic LLM settings, models are updated with information about existing topics rather than brand-new isolated topics, so examining whether this distinction matters for neural networks could offer interesting insights.
> > > > >
> > > > > **W4:**
> > > > > Thank you for the detailed justification. The discussion of planar vs. curved manifolds is helpful, and I agree that this is a promising direction for future research.
> > > > >
> > > > > **W5:**
> > > > > The definition in the paper could still be sharpened. The current phrasing, “tasks with low CKA when trained in isolation”, would also apply to the crossing task, which shows near-zero CKA. However, as far as I understand, you describe a task as “divergent” when it is sabotaging multi-task training.
> > > > >
> > > > > **Q2:**
> > > > > The expanded linear-probe analysis (Fig. 8 and Fig. 6) is helpful and addresses my question well. While several tasks appear not fully converged within the shared training budget, I understand the authors’ reasoning for keeping training lengths consistent, and I do not expect the qualitative conclusions to change with longer training.
> > > > >
> > > > >
> > > > > Overall, the rebuttal substantially improves the scope of the work. Some aspects could still be refined, but the experimental framework is compelling and provides a strong foundation for future research. In light of the new results and clarifications, I am leaning more toward acceptance and have adjusted my score accordingly.

---

> > > > > > ### Author Response · Authors · 2025-11-28
> > > > > >
> > > > > > Given the discussions will close soon, here are our replies:
> > > > > >
> > > > > > W1: Thank you for the praise. We indeed do the most accurate version (the former), which includes both seed variability and task variability.
> > > > > >
> > > > > > W2: Thank you for the appreciation! We totally agree [1000,9999] would have been an even cleaner choice.
> > > > > >
> > > > > > W3: The scattered Atlantis experiments concluded. I can say that all findings generalize as expected. Unfortunately, seems like we won't have time to ship them to you before the discussions close.
> > > > > >
> > > > > > W4: Thank you!
> > > > > >
> > > > > > W5: I see, this makes sense, we will define divergent tasks as tasks sabotaging multi-task fine-tuning, this way we don't overly claim the relation between pretraining CKA and fine-tuning gneralization is proven.
> > > > > >
> > > > > > ---
> > > > > >
> > > > > > Most importantly, the scattered Atlantis all checks out!
> > > > > >
> > > > > > ---
> > > > > > Thank you again for all your efforts!

---

### Official Review · Reviewer_PuHx · 2025-11-01

**Soundness:** 4
**Presentation:** 2
**Contribution:** 3
**Rating:** 6
**Confidence:** 4

**Summary:**

This paper performs a controlled analysis of how world models are formed in autoregressive models. 'World representations' are not fixed by the world itself but by the data‐generation tasks: single tasks carve up internal geometry very differently; training on diverse tasks forces these geometries to align ('Platonic' convergence).

The authors find:

-World representations emerge under autoregressive training: models first cluster nearby cities, then form a world‑aligned geometry; (x,y) becomes linearly decodable before task accuracy jumps.

-Single‑task training induces distinct geometries (e.g., distance yields a thread‑like structure, angle a 2‑D manifold)

-Multi‑task pretraining aligns representations: average CKA increases with task count (1→2→3), including between models that share no tasks

**Strengths:**

-The controlled setup for the experiments makes the results presented in this work convincingly support the idea of Platonic Representations presented in earlier work. By decoupling the world, data, and model, the authors control exactly what changes (tasks vs. world) and show models learn only from task outputs, never coordinates, so any alignment effects can be attributed to task diversity rather than data confounds.

-Setup is straightforward and training is possible at the level of academic resources. (i.e a 6‑layer, 128‑hidden, 4‑head Transformer with a 98‑symbol ASCII vocabulary, trained autoregressively). Fine‑tuning likewise uses manageable data.

-The addition of the Atlantis fine-tuning is a creative illustration of how downstream performance is impacted by the learned 'world representation' during pretraining. This provides a clean test of whether the 7‑task‑pretrained world manifold can absorb new entities and generalize them across tasks. Fine‑tuning on a single task with Atlantis yields a generalization matrix whose gains vary by task and correlate with pretraining CKA, directly tying the geometry learned during pretraining to downstream performance.

**Weaknesses:**

Since this paper focuses on analysis, most of the issues I encounter in this paper are focused on the clarity of presentation:

-Many figures are way too small (e.g. Figure 3-7), as a rule, the figure text should be sized similarly to the paper text

-Citations are missing in some parts of the paper (e.g. ? citations)

-Colors shown on World Map are not indicated with a legend.

Figure 8 is hard to tell the difference or what should be noticed in the contrast between the PCA/Linear probe subfigures for non-divergent vs divergent tasks. Are there quantitative measures that quantify how the new entities are different in the non-divergent vs divergent conditions?

If the goal is to demonstrate that the coordinates are placed in an orthogonal subspace and lie close to the origin, it would be more helpful to quantify it numerically rather than showing it visually.

Figure 8a is also hard to interpret. Without a clear way to interpret what the x-axis means, it's hard to understand what each entry in the matrix denotes and, consequently, what the vertical bands indicate.

**Questions:**

"This suggests that divergent tasks cause optimization to encode new entities in orthogonal subspaces rather than integrating them into the existing world manifold—explaining their failure to support cross-task generalization."
Can the authors make a statement (admittedly extrapolative) about how this is handled in the real-world by current models (e.g. LLMs) trained on data that may encode divergent tasks? Presumably, the data in the real world will not be as consistent as in the idealized setting posed in this paper.

"we do not claim that interventions to increase single-task CKA would necessarily improve fine-tuning generalization."
What are the author's thoughts on the intervention and how that would impact generalization? Was this intervention tried?

"Even excluding models with shared tasks, we find substantially higher CKA compared to single-task models" Can these pairs be isolated better from Figure 5c? Perhaps the matrix can be structured in a way where the partial overlap entries can be localized. I think non-overlapping tasks having higher alignment is an important result because it shows the common anchor is the World Map (i.e. the 'Platonic Space')

---

> ### Author Response · Authors · 2025-11-23
>
> ## **Response to Reviewer PuHx**
>
> Thank you for your thorough and constructive review. We are pleased that you found our controlled setup "convincingly support[s] the idea of Platonic Representations" and appreciated the creative use of Atlantis fine-tuning to study downstream effects of learned representations. Also thank you for appreciating the small setup approach, we're fans of tractable models where you can actually run all the experiments you want.
>
> Below we address each weakness and question.
>
> ---
>
> ### **W1: Small Figures**
>
> > *"Many figures are way too small (e.g. Figure 3-7), as a rule, the figure text should be sized similarly to the paper text"*
>
> Thank you for this feedback. we agree completely and have substantially revised **all** figures for readability, we didn't manage to match your gold standard but hopefully its at least much better!
>
> **Actions taken:**
> - Merged Figures 1, 2, and 3 into a single comprehensive framework and overview figure, eliminating redundancy while improving clarity
> - Resized all remaining figures with increased text size and concise presentation.
> - Significantly improved the layout and readability of Figure 6.
>
> ---
>
> ### **W2: Missing Citations / Missing World Map Legend**
>
> > *"Citations are missing in some parts of the paper (e.g. ? citations)"*
> > *"Colors shown on World Map are not indicated with a legend."*
>
> Thank you for catching these errors. Both issues have been fixed. We have also substantially expanded the related works section, as described in the general rebuttal.
>
> **Actions taken:**
> - Fixed all missing citation references
> - Added legend to the world map figure
> - Expanded related works in both main text and appendix (see general rebuttal)
>
> ---
>
> ### **W3: Quantitative Measure for Ill-Integration**
>
> > *"Figure 8 is hard to tell the difference or what should be noticed in the contrast between the PCA/Linear probe subfigures for non-divergent vs divergent tasks. Are there quantitative measures that quantify how the new entities are different in the non-divergent vs divergent conditions? If the goal is to demonstrate that the coordinates are placed in an orthogonal subspace and lie close to the origin, it would be more helpful to quantify it numerically rather than showing it visually."*
>
> Thank you for this suggestion. We recognize that the divergence from the world manifold is clearly visible in the interactive 3D visualizations we made but is difficult to see on paper. We have now added improved, close-up visualization and a linear probe reconstruction experiment which is easy to understand. We have also quantified mis-integration.
>
> In specific, we trained a linear probe on 4,000 non-Atlantis cities to predict (x, y) coordinates, then measured reconstruction error on Atlantis cities across all 84 fine-tuned models (4 seeds × 21 two-task combinations). Models fine-tuned with the distance task show approximately **5× higher Atlantis reconstruction error** compared to models fine-tuned without it (**Fig. 6d**). This confirms that divergent tasks cause new entities to be encoded in subspaces that do not align with the learned world manifold.
>
> **Actions taken:**
> - Added quantitative integration metric: linear probe reconstruction error on Atlantis cities (**Fig. 6d**)
> - Improved visual demonstration with better 3D PCA close-up views and linear probe reconstructions (**Fig. 6b**)
>
> ---
>
> ### **W4: Figure 8a Hard to Interpret**
>
> > *"Figure 8a is also hard to interpret. Without a clear way to interpret what the x-axis means, it's hard to understand what each entry in the matrix denotes and, consequently, what the vertical bands indicate."*
>
> We agree the original presentation was confusing. We have restructured this figure (now **Fig. 6**) for clarity. hopefully its much better now.
>
> **Actions taken:**
> - Transposed the matrix so that rows are models (fine-tuning task combinations) and columns are different evaluation tasks. (More conventional)
> - Moved the raw accuracy table to the appendix; the main figure now shows only the deviation from the expectation.
> - Added **Fig. 6c** which directly quantifies our main finding: task combinations including the distance task show substantially worse cross-task generalization
>
> **[1/n, Continued Below]**

---

> ### Author Response · Authors · 2025-11-23
>
> ---
>
> ### **Q1: Implications for Real-World LLMs**
>
> > *"Can the authors make a statement (admittedly extrapolative) about how this is handled in the real-world by current models (e.g. LLMs) trained on data that may encode divergent tasks? Presumably, the data in the real world will not be as consistent as in the idealized setting posed in this paper."*
>
> This is an great question! We address it from two angles:
>
> **During pretraining:** When Atlantis is included in the pretraining data, the model integrates it perfectly (App. Fig. 19). Reconstruction error for Atlantis cities falls within the distribution of regular cities (**Fig. 6d**, green line). This demonstrates that pretraining creates a window where joint representations form together, and the integration challenge is specific to post-hoc fine-tuning, not an artifact of Atlantis's geometry or location. This explains why LLMs, even if there might be divergent tasks lurking in the data, will probably have no problem integrating knowledge into a representation space.
>
> **During fine-tuning:** Our findings resonate with empirical observations in the LLM literature that fine-tuning often fails to trigger expected knowledge updates. The reversal curse [1] and challenges in integrating new factual knowledge [2] both suggest that gradient-based adaptation sometimes struggles to properly integrate new information into existing representational structures, similarly to our observation that divergent tasks encode new entities in hidden subspaces.
>
> **Actions taken:**
> - Added pretraining-with-Atlantis experiment (1 model) demonstrating successful integration when Atlantis is present from the start (**Fig. 6d**, green line + App. Fig. 19)
>
> ---
>
> ### **Q2: Interventions to Increase Single-Task CKA**
>
> > *"What are the author's thoughts on the intervention and how that would impact generalization? Was this intervention tried?"*
>
> We did not attempt interventions to increase single-task CKA. Our finding is purely correlational at this stage: we observe that single-task CKA predicts fine-tuning generalization, but we do not claim a causal relationship.
>
> Our implicit hypothesis is that the failure to properly integrate new entities during fine-tuning reflects an interaction between the model architecture and the data/task structure and this interaction affects both pretraining runs on single-tasks and fine-tuning dynamics in similar ways. Testing causal interventions (e.g., regularization toward higher CKA during pretraining) is an interesting (but seems very challenging) direction for future work, but beyond the scope of our current study.
>
> ---
>
> ### **Q3: Isolating Non-Overlapping Task Pairs in CKA Matrix**
>
> > *"Can these pairs be isolated better from Figure 5c? Perhaps the matrix can be structured in a way where the partial overlap entries can be localized. I think non-overlapping tasks having higher alignment is an important result because it shows the common anchor is the World Map (i.e. the 'Platonic Space')"*
>
> We agree this is an important result, glad you share this! Unfortunately, there is no matrix row/col ordering that will cleanly separates overlapping from non-overlapping pairs.
>
> However, we have taken steps to emphasize this finding:
>
> **Actions taken:**
> - Added explicit annotation in the figure clarifying that CKA values reported for multi-task comparisons exclude partially overlapping task pairs
>
> ---
>
> **Summary**
>
> Thank you again for the thoughtful feedback! Our paper's presentation improved substantially through this revision. We hope our changes and new experiments address your concerns adequately, and would be glad if our paper can be recommended more strongly.
>
> We are happy to discuss further or address any remaining questions.
>
> ---
>
> **References:**
>
> [1] Berglund et al. (2024): https://arxiv.org/abs/2309.12288
>
> [2] Lampinen et al. (2025): https://arxiv.org/abs/2505.00661v3

---

> > ### Comment · Reviewer_PuHx · 2025-11-25
> >
> > Thanks for the responses. Looking at the updated paper, I think my points in my original review have been addressed. This paper represents a very nice intuitive setting where we would expect world representations and posses a high degree of controllability for the introduction of counterfactuals (e.g. Atlantis). Much of the value in this paper is in simplicity of what a world model means.
> >
> > Despite other reviewers mentioning the impact being unclear as the results are not surprising, I lean more towards accept and increase my score.

---

> > > ### Author Response · Authors · 2025-11-28
> > >
> > > Thank you very much for your praise!
> > >
> > > One small comment: Atlantis is intentionally a "consistent update" instead of a "counterfactual"(which is often something which is in conflict with the pretraining knowledge).
> > >
> > > In other words, it is simply a non-disruptive addition to the world. This was motivated by Hase et al 2024 (https://arxiv.org/abs/2406.19354 ) since if the edit confuses the pretraining world, then it is unclear how to define generalization.
> > >
> > > ----
> > >
> > > Thank you again and let us know if you have any other questions!

---

### Author Response · Authors · 2025-11-23

Dear Reviewers,

We genuinely thank you for all the feedback. In a world with many LLM-generated reviews, we are grateful to have received thoughtful reviews from reviewers who clearly read, understood, and engaged with our work.

Reviewers found our setup **"convincingly support[s] the idea of Platonic Representations"** (PuHx), the framework **"well-designed"** and **"elegant"** (8dPS), experiments **"simple and diagnostic"** with **"solid evidence"** (oLe1), and results shown **"very cleanly"** (taAU). All four reviewers accurately summarized our three main findings—emergence of world representations, multi-task convergence, and divergent task interference; suggesting the presentation was effective despite concerns about figure sizing.

The reviewers raised three primary concerns: (1) **single-seed experiments** limiting confidence in robustness, (2) **presentation issues** including small figures and missing quantitative metrics, and (3) **calibration of PRH claims**. Motivated by your suggestions, we ran extensive additional experiments and substantially revised the paper:

---

**New Experiments (272 models, 13 new appendix figures):**
1. **Multi-seed pretraining (63 models):** 3 seeds × 21 task combinations to validate representational patterns across initializations.
2. **Exhaustive task combinations (63 models):** All C(7,1)+C(7,2)+C(7,3) = 63 combinations, confirming CKA convergence holds over all task pairs.
3. **Multi-seed fine-tuning (116 models):** 4 seeds × 29 fine-tuning configurations. All generalization matrices now report mean ± SEM.
4. **Pretraining with Atlantis (1 model):** Atlantis integrates perfectly when included from pretraining, confirming integration failure is a fine-tuning phenomenon, not geometric artifact (**Fig. 6d**, green line; App. Fig. 19).
5. **Wider model (29 models):** 2× hidden size; divergent task pattern persists on a bigger model (**App. Fig. 20**).
6. **Quantitative integration metric:** Linear probe reconstruction on 84 FT2 models; distance-containing models show ~5× higher Atlantis reconstruction error (**Fig. 6d**).

---

**Major Paper Revisions:**
- **Figures:** Merged Figs 1-3 into single overview; resized all figures with readable text; added new quantitative analysis (**Fig. 6b,c,d**).
- **PRH claims:** Clarified our evidence supports the *Multitask Scaling Hypothesis* (one proposed mechanism for PRH), not full cross-architecture/cross-modality convergence; added explicit limitation in **Results** and **Discussion**.
- **Related works:** Substantially expanded main text and added dedicated multi-task learning section in **Appendix**.
- **Introduction:** Shortened and focused to directly connect to research questions (per oLe1).
- **Appendix:** Added 13 new figures showing training dynamics, per-seed CKA matrices, per-seed generalization results.
- **Minor issues:** Fixed missing citations, world map legend, figure caption errors, abbreviation definitions.

---

**Key Findings from Multi-Seed Experiments:**

We ran 3 seeds for all pretraining experiments and quantified variance. This strengthened our findings:

- **Core results replicate:** Distance task consistently most divergent across all seeds; all fine-tuning generalization patterns hold (**Fig. 2b, 3c-d, 5a-b, 6a,c,d**; **App. Figs. 11-18**).
- **Multi-task reduces variance:** Increasing task count increases both cross-task CKA (alignment with models trained on different tasks) *and* same-task-different-seed CKA (alignment across initializations) (**App. Fig. 14b**).
- **Seed-averaging improves correlation:** CKA-to-generalization R² improves 0.126 → 0.188, suggesting single-seed measurements were noisy estimates (**Fig. 5b**).

We now report mean ± SEM for all CKA and show all individual seed results in Appendix.

---

We thank the reviewers for feedback that greatly improved our work. We address individual concerns below.

---

### Meta-Review · Area_Chair_J4EL · 2025-12-18

**Summary:**

Reviewer PuHx mainly concerned the clarity of presentation. Reviewer 8dPS focused on the conclusion consistency under different random seeds and argued that character-based city encoding is not enough for the validation of platonic representation and generalization. Reviewer oLe1 pointed out the topic in introduction section is too general on representation learning but the paper focuses on only city representations. Reviewer taAU argued that the evidence for platonic representation hypothesis is somewhat weak and not verified on different architectures.

**Reviewer Concerns:**

The concern about the validation on platonic representation hypothesis is not well addressed. The AC argues that authors can present more results on more LLM architectures, including open source LLMs to verify platonic representation hypothesis.

**Reviewer Scores:**

Reviewer PuHx: 6;

Reviewer 8dPS: 4;

Reviewer oLe1: 6;

Reviewer taAU: 2

---

### Decision · Program_Chairs · 2026-01-26

Reject